# Error Analysis of Non-Destructive Ultrasonic Testing of Glass Fiber-Reinforced Polymer Hull Plates

**Zhiqiang Han** [1] , **Jaewon Jang** [1] , **Sang-Gyu Lee** [1] , **Dongkun Lee** [2] **and Daekyun Oh** [2],*

[1] Department of Ocean System Engineering, Mokpo National Maritime University, Mokpo 58628, Korea; hzq910413@gmail.com (Z.H.); wodnjs1910@naver.com (J.J.); lsg93630@gmail.com (S.-G.L.)
[2] Department of Naval Architecture and Ocean Engineering, Mokpo National Maritime University, Mokpo 58628, Korea; dklee@mmu.ac.kr
* Correspondence: dkoh@mmu.ac.kr; Tel.: +82-61-240-7238

**Abstract:** Glass fiber-reinforced polymer (GFRP) ship structures are generally fabricated by hand lay-up; thus, the environmental factors and worker proficiency influence the fabrication process and presence of error in the non-destructive evaluation results. In this study, the ultrasonic testing of GFRP hull plate prototypes was conducted to investigate the statistical significance of the influences of the design parameters, e.g., the glass fiber weight fraction (Gc) and thickness variations, on the measurement error. The GFRP hull plate prototypes were fitted with E-glass fiber chopped strand mats (40 wt % content) with different thicknesses (7.72 mm, 14.63 mm, and 18.24 mm). The errors in the thickness measurements were investigated by conducting pulse-echo ultrasonic A-scan. The thickness variation resulted in increased error. Furthermore, hull plate burn-off tests were conducted to investigate the fabrication qualities. Defects such as voids did not have a significant influence on the results. The statistical analysis of the measurement errors confirmed that the thickness variations resulted in a strong ultrasonic interference between the hull plates, although the hull plates had similar specific gravity values. Therefore, the ultrasonic interference of the layer group interface should be considered to decrease the GFRP hull NDE errors with respect to an increase in the thickness and Gc.

**Keywords:** non-destructive testing; small craft; hull plate; glass fiber-reinforced polymer; ultrasound; thickness measurement errors

## 1. Introduction

The measurement of the thickness of a constructed hull via non-destructive testing (NDT) is a fundamental step in the inspection process. The analysis of the results allows for an inspector to determine whether the local or longitudinal strength of the hull structure meets the marine classification requirements. Furthermore, the presence of defects such as cracks on the hull plate can be identified via NDT [1,2].

NDT methods are widely used for hulls made of steel. However, small ships such as fishing and recreational boats made of glass fiber-reinforced polymers (GFRPs) have relatively large design margins [3], and visual inspection is the most common NDT method [4]. During the construction of a GFRP hull plate, it is challenging to fabricate the hull plate according to the required thickness, given that the fabrication process is based on the hand lay-up method using two materials, namely, glass fibers and resins. In addition to worker proficiency, environmental factors such as temperature and humidity can influence the hull plate thickness [5].

Examples of the most common NDT methods include X-ray inspection, thermography, and ultrasonic inspection, which are widely used for testing various composite structures [6]. However, for application to hull structures, the NDT method should be portable. Therefore, several NDT methods are ineffective and unreliable for the measurement of thick GFRP hull plates [7]. Among the various non-destructive examination methods,

pulse-echo ultrasonic testing is the most practical approach [8], as it requires measurement from only one side, which is particularly suitable for the shape of a ship hull.

However, there are numerous challenges associated with the ultrasonic NDT of a GFRP hull structure. Metallic materials such as steel have an internal structure that is relatively homogeneous. In contrast, GFRP materials are manufactured by blending two different materials, namely, glass fibers and resins. Thus, they have inhomogeneous properties, unlike steel. Glass fiber-reinforced polymers have a lower specific gravity than steel and exhibit increased scattering or absorption of ultrasonic waves due to irregularities in the crystal grains (Figure 1) [7]. Furthermore, the specific gravity of the plate changes depending on the glass fiber weight fraction, i.e., glass content (Gc), which is the weight ratio between the glass fiber and resin. Thus, the utilization of ultrasonic NDT for the evaluation of composites is challenging [9,10]. Additionally, inner defects such as voids may be present in composite materials, which can reduce the density and cause lower mechanical properties [11,12] and inhibit ultrasonic wave propagation (Figure 1).

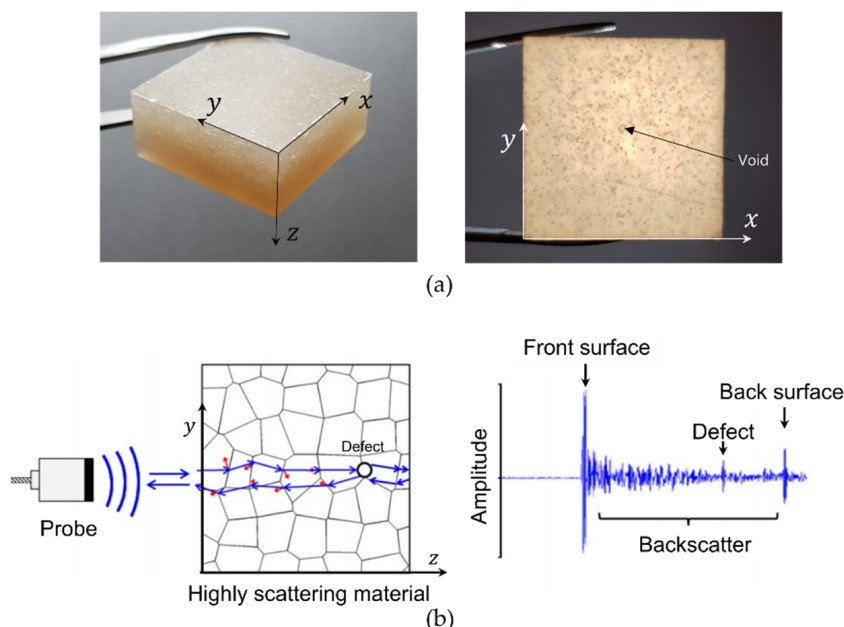

**Figure 1.** Ultrasonic NDT of composites: (**a**) sample of GFRP hull plate, and (**b**) high ultrasonic attenuation due to inhomogeneous properties of composites; reprinted with permission from ref. [10]. 2015 Van Pamel.

The use of ultrasonic NDT methods for the examination of GFRP hull structures was investigated in previous studies. In a recent study conducted by Lee et al. [13], the influences of the fabrication characteristics of a GFRP hull structure on the ultrasonic NDT results were investigated. Changes in the Gc of the hull plates were confirmed to influence the accuracy of the ultrasonic NDT results. In particular, the influence was more significant for plates with relatively high Gc values (higher than 50%) when compared with those with commercial-level Gc values (30–50%). In another study conducted by Lee et al. [14], higher measurement error rates were observed for hull plates with internal defects such as voids, even if they were designed using the same Gc values and lamination schedules.

Glass fiber-reinforced polymer hull plates generally exhibit large thickness variations according to the size of the structure and their applications. Based on the results of the aforementioned studies, in this study, the influences of changes in the plate thicknesses in the hull plates of fishing boats, which are the majority of GFRP ships, on the ultrasonic thickness measurement errors were investigated. In addition, the effects of inner defects that may occur during actual fabrication on measurement error were analyzed.

## 2. Materials and Methods

### 2.1. Ultrasonic Testing of Composite Materials

Ultrasonic NDT is commonly used in various fields. Ultrasonic waves can be employed for material thickness determination and defect identification. This is realized by analyzing the pulse-echo velocity and time required for the propagation of the ultrasonic waves generated from the probe through the sample from the front to the back surface, or to an internal defect, after reflection. These data were analyzed to determine the sample thickness, in addition to the location and size of the defects [15]. Thus, the determination of an appropriate wave propagation speed or ultrasonic pulse-echo velocity of a given material is critical and dependent on the physical properties of the sample, such as its density.

For homogeneous materials, including steel, the ultrasonic pulse-echo velocity and impedance for a given material density are available for review [13] and can thus be applied to testing. In a homogeneous material, the ultrasonic propagation speed is constant, and the propagation characteristics are consistent. Therefore, the ultrasonic NDT of homogeneous materials such as steel yields relatively accurate results and is simple to conduct. However, there are no existing data on the ultrasonic pulse-echo velocity for GFRP composite materials. Moreover, the fiber texture of the GFRP material is inconsistent, and the composite material is not structurally uniform due to the difference between the density of the fibers and resins. Given that the propagation characteristics of ultrasonic waves are complex in composite materials, it is difficult to establish accurate inspection conditions.

The GFRP hull plate is typically thicker than the composites used for automobile and aviation applications. Given that GFRP composites are structurally inhomogeneous, the scattering and absorption of ultrasonic waves generally occur in these materials (Figure 1). Accordingly, their large thickness significantly influences the accuracy of the NDT results. In general, the thickness of GFRP ship structures ranges from 5 to 10 mm, and in the cases of special purpose ships such as naval vessels, it can range from 10 to 15 mm [16]. The hull plate thickness of a ship is typically approximately 10 mm [3,17], whereas the hull plate of a yacht with a length of 15 has a thickness of approximately 15 mm [18]. The parameters that influence the GFRP hull plate thickness include the ship design elements such as the ship structure size, ship speed, and its mechanical properties, depending on the material design conditions. The mechanical properties of the hull plates are determined by the Gc or the weight percentages of the fiber and resin, and they are very important design variables [3,18–21]. The small ship hull plates that are typically observed in fishing and recreational boats are generally produced via the hand lay-up method, and they generally have Gc values of 30–50% [22]. Given that the Gc is a critical factor that determines the specific gravity of GFRP hull plates, it is directly related to the ultrasonic pulse-echo velocity. There are no ultrasonic pulse-echo velocity data for GFRP materials; therefore, they should be determined based on the properties of the hull plates.

Furthermore, internal defects such as voids can be introduced due to the fabrication process of the composite material. In previous studies conducted by Stone and Clarke [23], Ishii et al. [24], and Lin et al. [25], voids were confirmed to disrupt the propagation of ultrasonic waves and increase scattering and absorption. In addition, Lee et al. [13] observed that a void content of 5% in GFRP hull plates considerably reduced the ultrasonic pulse-echo velocity. With an increase in the Gc of plates, there is an increase in the probability of defect formation, i.e., voids [13,26], and these defects lead to the disruption of ultrasonic wave propagation. Thus, voids can introduce errors into the thickness measurements. To improve the accuracy of the NDT results for application to GFRP hull plates, the use of a low-frequency probe in the range of 0.50–2.25 MHz is recommended [13,14,27].

In this study, the ultrasonic non-destructive examination of GFRP hull plates was conducted, and the thickness measurement errors were analyzed, while considering the design conditions and fabrication properties. Table 1 summarizes the properties of a typical GFRP hull plate and its influence on the ultrasonic acoustic properties. Accordingly, in this study, the prototypes of the plates were utilized to evaluate and analyze the thickness measurement errors.

**Table 1.** The GFRP hull plate properties and ultrasonic inspection set conditions.

| Properties of GFRP Hull Plate | | Changes in the Ultrasonic Acoustic Properties | Set Conditions |
| --- | --- | --- | --- |
| Design | Low Gc (30–50%)/density ($\rho$) | Changes in specific gravity $\rightarrow$ Ultrasonic pulse-echo velocity amorphous characteristics | Ultrasonic pulse-echo velocity |
| | Increased thickness and ply structures | Ultrasonic wave scattering and absorption | Low frequency |
| Fabrication | Hand lay-up method; defect formation during fabrication | Disruption of ultrasonic waves Pulse-echo velocity and attenuation (damping) changes | - |

## 2.2. Research Method

In previous studies [13,14], the cause of the large error associated with the ultrasonic NDT of GFRP hull plates was investigated to improve the accuracy of the results. It was determined that the Gc of the hull plates and defects such as voids play a significant role in the observed errors. Based on previous studies, ultrasonic NDT was conducted in this study on hull plates with design parameters similar to those of GFRP fishing ships, to obtain thickness measurements, and to perform statistical analysis on the thickness measurement errors.

To perform ultrasonic NDT on the hull plates, ultrasonic pulse-echo velocity data are required with respect to changes in the Gc. We fabricated hull plates with thicknesses of 7.50–10.00 mm and Gc values of 30%, 40%, and 45%, which is within the weight fraction range for typical hull plates (30–50%) fabricated via the hand lay-up method, as reported in a previous study. In this study, the changes in the ultrasonic pulse-echo velocity according to the Gc were determined based on the hull plate used in previous studies.

To investigate thickness measurement errors, the ultrasonic pulse-echo velocity results were obtained for three hull plates used in actual fishing ships with the same design Gc of 40% and three different thicknesses (8 mm, 15 mm, and 20 mm). Moreover, amplitude scan ultrasound biometry (A-scan) results were obtained using a probe with a diameter of 12.70 mm and frequency of 1.00 MHz for the pulse-echo ultrasonic NDT of the plates. The obtained thickness values were compared with the values obtained using a Vernier caliper to calculate the measurement errors. Given that the thickness measurement errors are influenced by the hull plate thickness and internal defects such as voids, statistical techniques were used to examine the influence of each variable on the measurement errors.

## 3. Prototypes of GFRP Hull Plates

### 3.1. Design and Fabrication of GFRP Hull Plate Prototypes

Generally, the majority of GFRP ships are fishing ships with gross tonnages (GTs) of 5–10 tons; with hull plate thicknesses of approximately 10 mm and keel thicknesses of 20 mm [28]. These hull plates have Gc values in the range of 30–50%. To determine the influence of the hull plate thickness on the ultrasonic NDT results, the hull plates in this study were fabricated by considering the thickness range of general fishing ships with GTs of 5–10 tons.

To design and fabricate the hull plates, a 450 g/m$^2$ E-glass fiber chopped strand mat (CSM) was used for reinforcement, and polyester resin was used as the matrix material. The relative density of each material for the analysis of the experimental results was measured using a hydrometer (Alfa Mirage Co., Ltd., Osaka, Japan; EW-300SG) based on the ASTM D792 standard [29]. The density of the E-glass fiber was measured as 2.62 g/cm$^3$, whereas that of the polyester resin was 1.23 g/cm$^3$. Table 2 presents the design parameters of three hull plates that were designed and fabricated using the hand lay-up method.

**Table 2.** Design properties of the hull plates used for thickness measurement error determination.

| Item | Application 1 | Application 2 | Application 3 |
|---|---|---|---|
| Design Gc (%) | 40% | 40% | 40% |
| lamination schedule | CSM × 10 | CSM × 21 | CSM × 26 |
| Design thickness (mm) | 7.40 | 15.54 | 19.24 |

*3.2. Quality Control of the GFRP Hull Plate*

In previous studies, it was reported that the Gc and defects such as voids have significant influences on the A-scan results. Therefore, in this study, the average prototype thickness, Gc, and void content were measured for the three hull plate prototypes. In each case, considering the diameter of the probe and the size of the prepared hull plates, 30 measurement locations (2 cm × 2 cm) were defined. The thickness of each hull plate prototype was measured using a vernier caliper (HANDO Digital caliper, M500-182M), and average thicknesses of 7.72 mm (Application 1), 14.63 mm (Application 2), and 18.24 mm (Application 3) (Figure 2) were obtained.

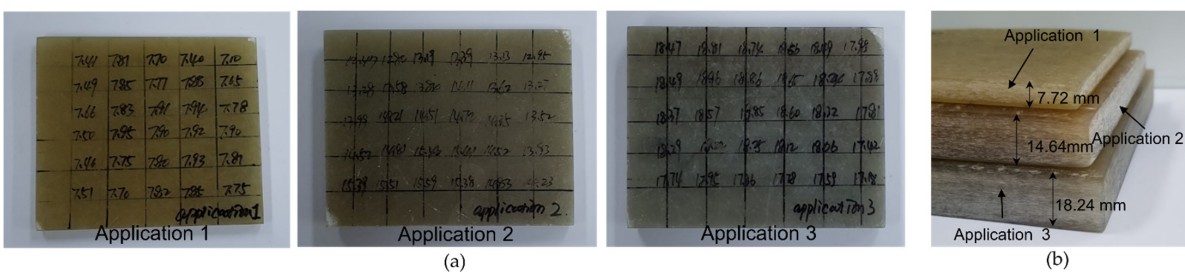

(a)                                                                                                    (b)

**Figure 2.** Thickness measurement for each hull plate prototype: (**a**) top view and (**b**) isometric view.

A burn-off test was performed to measure the Gc values and void contents [18]. From the experimental hull plate, prototypes with dimensions size of 2 cm × 3 cm were obtained. According to the ASTM D3171 standard [30], five prototypes for each hull plate or 15 prototypes in total were used for the burn-off test. An electric muffle furnace with a capacity of 4.5 L was preheated to approximately 600 °C and used for the burn-off test. The heating and cooling of the prototypes were repeated to burn the resins until the prototype mass was <0.001 g. The remaining glass fiber was analyzed to obtain the Gc values and void contents (Tables 3 and 4). As can be observed from Table 3, the average Gc values of the three hull plates were 40.67% (Application 1), 40.29% (Application 2), and 43.83% (Application 3), and a minimal deviation from the designed Gc (40%) was observed due to the fabrication characteristics. According to Table 4, the void content was approximately 0.92–1.80%, which is comparable to the void content of a typical hull plate [27].

**Table 3.** Measured Gc of the fabricated hull plate prototypes.

| Type | Design Gc (%) | Thickness (mm) | Measured Gc (%) | | | | | Average (%) |
|---|---|---|---|---|---|---|---|---|
| Application 1 | 40.00 | 7.72 | 41.58 | 40.01 | 39.59 | 40.27 | 41.87 | 40.67 |
| Application 2 | 40.00 | 14.63 | 43.00 | 40.18 | 39.36 | 38.73 | 40.19 | 40.29 |
| Application 3 | 40.00 | 18.24 | 42.69 | 44.02 | 42.39 | 45.78 | 44.28 | 43.83 |

**Table 4.** Measured void content of the fabricated hull plate prototypes.

| Type | Design Gc (%) | Thickness (mm) | Measured Void Content (%) | | | | | Average (%) |
|---|---|---|---|---|---|---|---|---|
| Application 1 | 40.00 | 7.72 | 1.15 | 0.73 | 1.09 | 0.91 | 0.71 | 0.92 |
| Application 2 | 40.00 | 14.63 | 1.48 | 0.85 | 0.94 | 1.17 | 1.50 | 1.19 |
| Application 3 | 40.00 | 18.24 | 1.89 | 1.55 | 1.69 | 2.13 | 1.72 | 1.80 |

## 4. A-Scan of the GFRP Hull Plates

### 4.1. Determination of the Pulse-Echo Velocity of an Ultrasonic A-Scan

To perform an ultrasonic A-scan on the hull plate prototypes, an appropriate ultrasonic pulse-echo velocity should be determined for each Gc. In a previous study [31], the ultrasonic pulse-echo velocity was obtained using an E-glass fiber CSM and polyester resin composite hull plate prototypes for Gc values of 30%, 40%, and 45%. The average thickness values of the hull plate prototypes were 9.87 mm, 8.71 mm, and 7.56 mm; the void content, as determined in a previous study, was in the range of 0.84–1.48% based on the burn-off test results. The prototypes had similar qualities as the hull plates [31].

To measure the ultrasonic pulse-echo velocity, measurement locations were set for the three hull plates with a width and height of 2 cm. This resulted in 50 measurement locations per plate; thus, a total of 150 measurements were obtained using a Vernier caliper and an ultrasonic thickness measuring instrument to determine the ultrasonic pulse-echo velocity. The resulting measurements with respect to the changes in Gc are shown in Figure 3.

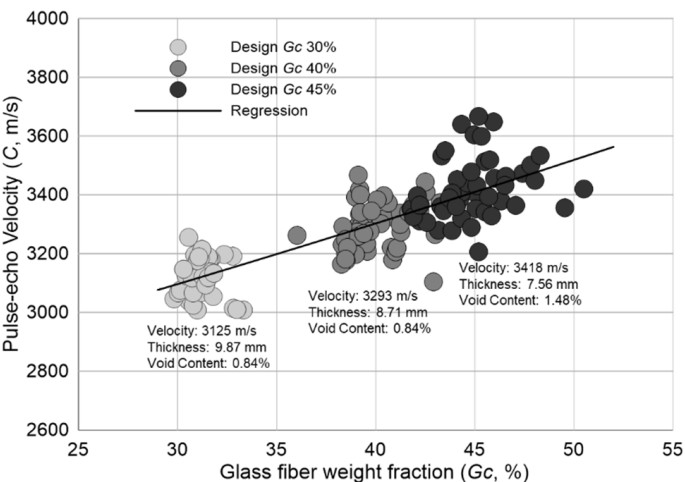

**Figure 3.** Statistical results for pulse-echo velocity as a function of Gc.

As shown in Figure 3, the ultrasonic pulse-echo velocities of the three hull plate prototypes with Gc values of 30%, 40%, and 45% were 3125 m/s, 3293 m/s, and 3418 m/s, respectively. The results revealed an increase in the pulse-echo velocity with an increase in the Gc. This can be attributed to the increase in the relative density with an increase in the Gc. In composite materials, the ultrasonic pulse-echo velocity is determined by the material density. These results are similar to those of a previous study [13].

Using regression analysis, the relationship between the ultrasonic pulse-echo velocity and typical Gc changes (30–50%) of GFRP hull plates was established. Consequently, a regression model was obtained, as expressed by Equation (1).

$$C = 0.05Gc^2 + 17.29Gc + 2534.31; R^2 = 0.67 \tag{1}$$

### 4.2. Thickness Measurement Using Ultrasonic Waves and Error Comparison Analysis

An appropriate ultrasonic pulse-echo velocity should be determined before ultrasonic instrumentation can be used for the determination of the hull plate prototype thicknesses. Hence, the average Gc (Table 3) from the burn-off test was used with Equation (1) to calculate the appropriate pulse-echo velocity. The results are shown in Table 5. Given that the hull plates were fabricated using the hand lay-up method, there was a minimal deviation from the design Gc, which is a common phenomenon during the hull plate fabrication process [13,18,31].

**Table 5.** Ultrasonic pulse-echo velocity implemented for hull plate prototypes.

| Items | Application 1 | Application 2 | Application 3 |
|---|---|---|---|
| Average Gc (%) by burn-off test | 40.67 | 40.29 | 43.83 |
| Velocity (C, m/s) | 3320 | 3312 | 3379 |

The thickness measurement errors were calculated by comparing the thickness measurements obtained via ultrasonic testing with those obtained using a Vernier caliper. The comparison results are shown in Figure 4, which indicate that the values obtained using both methods were comparable for Application 1. However, the two thickness measurements differed considerably for Applications 2 and 3, and the measured values obtained using the ultrasonic test were greater than the actual thickness values. In addition, a large discrepancy among the thickness measurements obtained using the ultrasonic test was observed for the thick hull plate prototypes (Figure 4a).

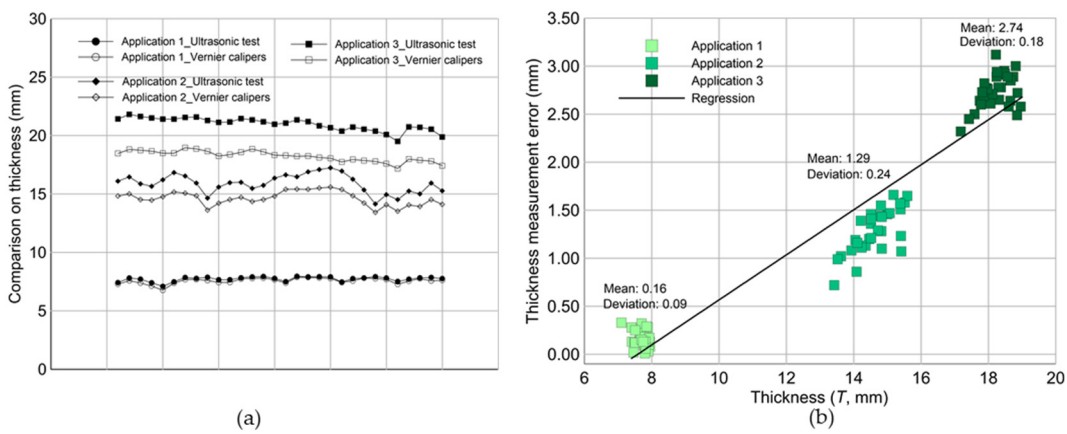

**Figure 4.** (**a**). Comparison of actual thickness and thickness determined using ultrasonic testing for each hull plate prototype, and (**b**) comparison of measured thickness errors as a function of the hull plate prototype thickness.

The thickness measurement errors were determined for Applications 1, 2, and 3 based on all the measurement locations. The results indicated that for the hull plates with similar Gc values, there was an increase in the thickness measurement error in accordance with an increase in the thickness; the average measurement errors were 0.16 mm (2.07%), 1.29 mm (9.18%), and 2.74 mm (14.92%) for Applications 1, 2, and 3, respectively (Figure 4b). Furthermore, the standard deviations for Applications 2 and 3 were greater than that of Application 1 by factors of 2.7 and 2, respectively.

## 5. Results and Discussions

As previously mentioned, there was an increase in the thickness measurement error with an increase in the thickness of the hull plate prototypes, as can be seen from the measurement results for Applications 1, 2, and 3. For Application 1, the measurement error was approximately 2%; whereas Applications 2 and 3 had errors of 10% and 15%, respectively. As presented in this section, the causes of the thickness measurement errors were investigated, in addition to the relationship between the measurement error and sample thickness.

### 5.1. Comparative Analysis of the Ultrasonic Pulse-Echo Velocities of the GFRP Hull Plate Prototypes

To identify the causes of the observed thickness errors, the ultrasonic pulse-echo velocity used for each measurement location was determined for Applications 1, 2, and 3. Figure 5 illustrates the ultrasonic pulse-echo velocity used for each measurement location and statistical analysis results for the ultrasonic pulse-echo velocity as a function of Gc.

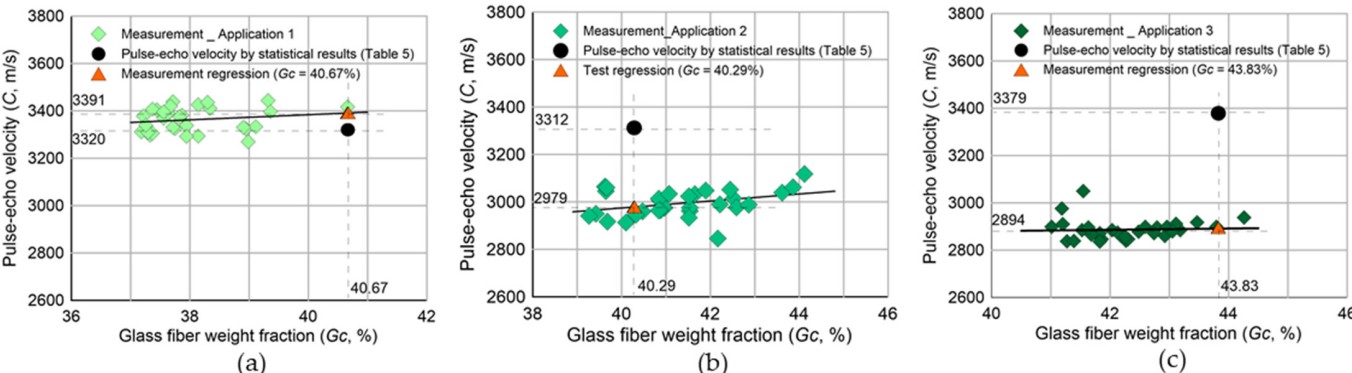

**Figure 5.** Comparison of ultrasonic pulse-echo velocity for Application (**a**) 1, (**b**) 2, and (**c**) 3.

The thickness of the hull plate used for determining the ultrasonic pulse-echo velocity as a function of the Gc was approximately 7.50–10.0 mm (Figure 3), which is similar to the hull plate prototype thickness in Application 1 (7.72 mm). The average ultrasonic pulse-echo velocity was 3391 m/s for Application 1, which is similar to the ultrasonic result (3320 m/s) obtained as a function of the changes in the Gc (in Table 5). However, for Applications 2 and 3, the differences between the ultrasonic pulse-echo velocities were 333 m/s and 485 m/s respectively, which exhibited greater velocity discrepancies when compared with Application 1. With an increase in the thickness of the hull plate prototype, there was an increase in the discrepancy between the actual ultrasonic pulse-echo velocity used to determine the actual thickness and statistical ultrasonic pulse-echo velocity.

With an increase in the GFRP hull plate thickness, there was an increase in the resin content, an increase in the number of fabric plies, and a larger interface change between each ply, which led to a significant increase in the attenuation of the ultrasonic waves [9,29]. Accordingly, the propagation of the ultrasonic waves was inhibited, and the ultrasonic propagation path was more irregular [9,27]. This can explain the observed thickness measurement errors of the hull plates for Applications 1, 2, and 3.

In addition, the results of previous studies indicated that the void content can influence the thickness measurement error. Therefore, in this study, the thickness and void content were identified as the two main variables that contributed to the thickness measurement errors of the hull plates.

### 5.2. Influence of Void Content on Measurement Error

To evaluate the influence of the void content on the measurement error, the influence of the void content on the ultrasonic pulse-echo velocity was analyzed. According to Table 4, the void contents of the hull plates for Applications 1, 2, and 3 were approximately 0.92–1.80%. Furthermore, the ultrasonic pulse-echo velocity with respect to the hull plate void content was determined as 0.84–1.48%. Void contents within this range are typical of marine composites [27,31], and all the hull plates exhibited similar qualities. Furthermore, in a study conducted by Lee et al. [13], it was reported that a hull plate void content of 5% had a significant influence on the pulse-echo velocity, thus leading to a velocity decrease. Therefore, given that the GFRP hull plate had a void content of <2%, it was determined that the void content at this level did not contribute significantly to the thickness measurement error and was therefore excluded from further analysis.

### 5.3. Statistical Analysis of the Influence of Thickness Changes on the Measurement Error

A one-way analysis of variance (ANOVA) [32,33] was performed on the data to determine if there was a statistically significant relationship between the hull plate thickness and the measurement error for Applications 1, 2, and 3. Hypothesis testing was performed to examine the normality and homogeneity of the variance with respect to the thickness measurement error for each independent group. As illustrated in Figure 6, the distribution

of data points matches well with a theoretical distribution line, and the *p*-values corresponding to Applications 1, 2, and 3 were 0.08, 0.58, and 0.93 for a significance level of 0.05, thus satisfying the normality assumption. In contrast, as shown in Figure 7, the *p*-value was <0.0001, and the data for the three groups did not satisfy the homogeneity of the variance assumption when the significance level was 0.05.

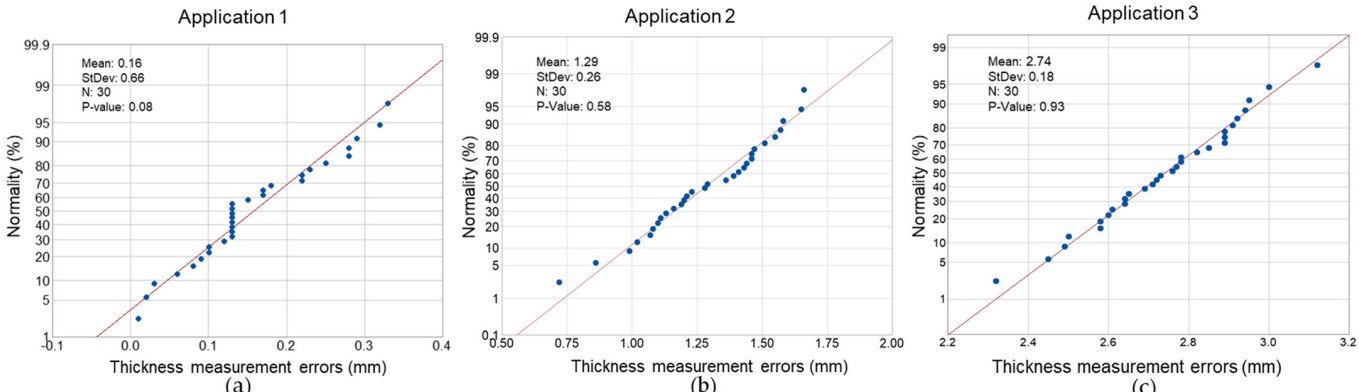

**Figure 6.** Testing the normality of the thickness measurement error data of Applications (**a**) 1, (**b**) 2, and (**c**) 3.

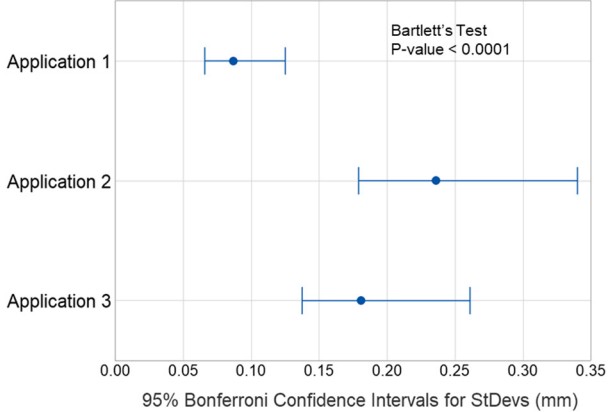

**Figure 7.** Testing the homogeneity of variance in the thickness measurement error data of Applications 1, 2, and 3.

Given that the thickness measurement errors for Applications 1, 2, and 3 did not satisfy the homogeneity of variance, Welch's test [34] was utilized, which is a more conservative method.

The average thickness measurement errors for Applications 1, 2, and 3 had a *p*-value close to 0 for a significance level of 0.05; hence, the null hypothesis was rejected, and there was a statistically significant difference between the three groups (Table 6).

Furthermore, a post-analysis was conducted based on the commonly used Games–Howell method [35] to determine which group contributed the most to the observed statistically significant difference. It was postulated that the observed difference is due to the smaller *p*-value with respect to the significance level (0.05) for the average of each group (Table 7a). All three groups exhibited significant differences (Table 7b).

Based on the statistical analysis results, the changes in the thicknesses of the hull plates for Applications 1, 2, and 3 had a statistically significant influence on the thickness measurement errors, and the changes in the plate thicknesses can significantly disrupt the propagation of ultrasonic waves.

Visual inspection is the most common NDT method for GFRP hull inspection. Hence, the findings of this study can therefore serve as a reference for GFRP hull quality in-

spection using the pulse-echo ultrasonic testing method while maintaining a relative low measurement error.

**Table 6.** The ANOVA results with the Welch's test.

| Source | Degrees of Freedom—Numerator | Degrees of Freedom—Denominator | F-Value | *p*-Value |
|---|---|---|---|---|
| (Thickness errors in Applications 1, 2, and 3) | 2 | 49.09 | 2554.13 | <0.0001 |

**Table 7.** (a) Games–Howell simultaneous tests for differences of means. (b) Grouping information using the Games–Howell method and 95% confidence.

| (a) | | | | | |
|---|---|---|---|---|---|
| Difference of Level | Difference of Means | SE | 95% CI | T-Value | *p*-Value |
| Application 1–Application 2 | 1.13 | 0.05 | (1.02,1.24) | 24.61 | <0.0001 |
| Application 1–Application 3 | 2.58 | 0.04 | (2.49, 2.67) | 70.53 | <0.0001 |
| Application 2–Application 3 | 1.45 | 0.05 | (1.32, 1.58) | 26.80 | <0.0001 |

| (b) | | |
|---|---|---|
| Factor | Mean | Grouping [a] |
| Thickness error in Application 1 | 0.16 | A |
| Thickness error in Application 2 | 1.29 | B |
| Thickness error in Application 3 | 2.74 | C |

[a] Means that do not share a letter are significantly different.

## 6. Conclusions

The purpose of this study was to investigate the influences of the design parameters of the GFRP hull plate to ultrasonic thickness measurement error. Based on an analysis of the pulse-echo ultrasonic A-scan error results of the GFRP hull plates, the main findings of this study were as follows.

- With an increase in the GFRP hull plate thickness, the resin content and interface change between the fiber and resin increase, thus leading to inaccurate thickness measurements due to the scattering and absorption of the ultrasonic waves.
- The void content of the GFRP hull plates contributes to the ultrasound thickness measurement error. However, in this study, the void content of the samples was approximately 2%, which had a minimal influence on the measurement error.
- According to previous studies, the Gc of GFRP hull plates should be considered as an important variable to improve the accuracy of ultrasonic NDT results. However, the results of this study demonstrated that the influence of the plate thickness on the measurement errors should be additionally considered during ultrasonic NDT.

**Author Contributions:** Conceptualization, D.O.; Methodology, D.O. and D.L.; Funding acquisition, D.O.; Manufacturing, Z.H. and J.J.; Test and investigation, Z.H. and S.-G.L.; Writing—original draft, Z.H. and J.J.; Writing—review and editing, D.O. All authors have read and agreed to the published version of the manuscript.

**Funding:** This work was supported in part by the 'IoT and AI-based development of Digital Twin for Block Assembly Process (20006978)' Program of the Korean Ministry of Trade, Industry and Energy, Republic of Korea.

**Institutional Review Board Statement:** Not applicable.

**Informed Consent Statement:** Not applicable.

**Data Availability Statement:** All data are presented in the paper.

**Conflicts of Interest:** The authors declare no conflict of interest.

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
