# Peer review of "Error Analysis of Non-Destructive Ultrasonic Testing of Glass Fiber-Reinforced Polymer Hull Plates"

_jcs, doi:10.3390/jcs5090238_

Round 1
Reviewer 1 Report
Overall, this is a good-quality manuscript. Few minor suggestions:
63: Use of sentences in direct format is not suggested. Like ‘We investigated..’ Use passive sentences.
Add a discussion on interactions or interaction check. This subject matter should be mentioned.
Some newer references could be added. Examples: https://doi.org/10.1016/j.compositesb.2019.107112 and https://doi.org/10.1007/s00170-020-05195-z
Make the conclusion in a paragraph. No need to be numbered.
Reviewer 2 Report
The submitted article “jcs-1367110-v1” entitled: “Error Analysis of Non-Destructive Ultrasonic Testing of Glass Fiber-Reinforced Polymer Hull Plates” evaluates the ultrasonic testing of Glass fiber-reinforced polymer (GFRP) hull-plate specimens with different thicknesses. The errors in the thickness measurements are measured and estimated using pulse-echo ultrasonic A-scan. Three GFRP hull-plate specimens have been designed and fabricated using the hand lay-up method and then tested. The paper falls within the scope of the Journal. The presented study is certainly of interest to the readers of the Journal, manuscript is well-structured and the findings obtained are of good quality. Figures are also helpful. Therefore, the paper could be accepted for publication after minor revision. The following comments and suggestions are raised for authors’ reference:
- In introduction, the literature review is rather informative but could be improved providing more convincing motivations of this research. Although the tasks and the research significance of the study are clearly defined, it is recommended to be highlighted. The main objectives of this work are also briefly stated and seem rather modest since extensive conclusion have been derived from this study.
- The relationship between the text and the figures could be improved; the text leads the readers to believe that the figures will provide the desired information and/or clarification of the work done but the figures do not really provide this. Discussion and further commentary of the figures would be helpful.
- Some additional comments and explanation concerning the selection of the width of the 20 mm mesh that leads to 30 measurement locations in each specimen could help the readers of the paper to comprehend this issue in-depth. Some clarifications are also required if a different mesh dimension selections (or else more or less measurements) could provide different results concerning the effectiveness of the test procedure.
